# Receptor for Advanced Glycation End Product, Organ Crosstalk, and Pathomechanism Targets for Comprehensive Molecular Therapeutics in Diabetic Ischemic Stroke

**DOI:** 10.3390/biom12111712

**Published:** 2022-11-18

**Authors:** Nivedita L. Rao, Greeshma B. Kotian, Jeevan K. Shetty, Bhaskara P. Shelley, Mackwin Kenwood Dmello, Eric C. Lobo, Suchetha Padar Shankar, Shellette D. Almeida, Saiqa R. Shah

**Affiliations:** 1Department of Biochemistry, Yenepoya Medical College, Yenepoya (deemed to be University), Mangalore 575018, Karnataka, India; 2Department of Biochemistry, School of Medicine, Royal College of Surgeons in Ireland Medical University of Bahrain, Muharraq 228, Bahrain; 3Department of Neurology, Yenepoya Medical College, Yenepoya (deemed to be University), Mangalore 575018, Karnataka, India; 4Department of Public Health, KS Hegde Medical Academy, Nitte (Deemed to be University), Mangalore 575018, Karnataka, India; 5College of Physiotherapy, Dayananda Sagar University, Bangalore 560111, Karnataka, India; 6School of Physiotherapy, D. Y. Patil (Deemed to be University), Navi Mumbai 400706, Maharashtra, India

**Keywords:** advanced glycation end products—AGEs, receptor for AGE (RAGE), high-mobility group box 1 (HMGB1) nuclear protein, Leukotriene B4, pathomechanisms, diabetes mellitus, ischemic stroke, cerebrovascular disease, therapeutic agents, comprehensive strategies

## Abstract

Diabetes mellitus, a well-established risk factor for stroke, is related to higher mortality and poorer outcomes following the stroke event. Advanced glycation end products(AGEs), their receptors RAGEs, other ligands, and several other processes contribute to the cerebrovascular pathomechanism interaction in the diabetes–ischemic stroke combination. Critical reappraisal of molecular targets and therapeutic agents to mitigate them is required to identify key elements for therapeutic interventions that may improve patient outcomes. This scoping review maps evidence on the key roles of AGEs, RAGEs, other ligands such as Leukotriene B4 (LTB4), High-mobility group box 1 (HMGB1) nuclear protein, brain–kidney–muscle crosstalk, alternate pathomechanisms in neurodegeneration, and cognitive decline related to diabetic ischemic stroke. RAGE, HMGB1, nitric oxide, and polyamine mechanisms are important therapeutic targets, inflicting common consequences of neuroinflammation and oxidative stress. Experimental findings on a number of existing–emerging therapeutic agents and natural compounds against key targets are promising. The lack of large clinical trials with adequate follow-up periods is a gap that requires addressing to validate the emerging therapeutic agents. Five therapeutic components, which include agents to mitigate the AGE–RAGE axis, improved biomarkers for risk stratification, better renal dysfunction management, adjunctive anti-inflammatory–antioxidant therapies, and innovative neuromuscular stimulation for rehabilitation, are identified. A comprehensive therapeutic strategy that features all the identified components is needed for outcome improvement in diabetic stroke patients.

## 1. Introduction

Diabetes mellitus, currently considered one of the largest epidemics, is associated with cardio-cerebrovascular complications. Together, they present a major public health burden in terms of healthcare economics, enhanced morbidity, and mortality, with the prevalence rising at an exponential rate worldwide [1,2]. A 26% increase in global stroke deaths was indicated during the past two decades [1]. Two-thirds of all strokes are estimated to occur in developing countries, which, despite their preventable nature, continue to occur increasingly [3,4]. Stroke-related deaths are expected to nearly double in North Africa and the Middle East by 2030 [4]. Stroke survivors experience motor impairments requisite to locomotion and post-stroke dysfunction of nearly 75%. Severe functional disability of 15–30% has been reported by the American Heart Association, which warrants more attention towards improving disease management strategies [5].

Stroke is defined as “Central nervous system (CNS) infarction is brain, spinal cord, or retinal cell death attributable to ischemia, based on 1. pathological, imaging, or other objective evidence of cerebral, spinal cord, or retinal focal ischemic injury in a defined vascular distribution; or 2. clinical evidence of cerebral, spinal cord, or retinal focal ischemic injury based on symptoms persisting ≥24 h or until death, and other etiologies excluded” [5]. Ischemic stroke due to thrombosis, embolism, or systemic hypoperfusion is one of the two broad categories of stroke, with the other being hemorrhagic stroke due to intracerebral hemorrhage or subarachnoid hemorrhage [5]. Strokes due to ischemic cerebral infarction comprise approximately 80% of strokes [6]. The ischemic cascade of activated biochemical reactions initiated in the brain and other aerobic tissues during and after ischemia causes neuronal death and usually lasts for 2-3 h but can last for days, even after normal blood flow returns [7]. The Trial of Org 10,172 in Acute Stroke Treatment (TOAST) system guides proper management decisions and categorizes ischemic stroke subtypes [8].

Diabetes mellitus, a metabolic disease, is also a major risk factor for cardiovascular and ischemic stroke disease development [9,10,11]. Enhanced generation and accumulation of advanced glycation end products (AGEs) occurring due to chronic hyperglycemia of diabetes has been implicated in the pathogenesis of macrovascular and microvascular complications leading to end-organ damage and cardio- and cerebrovascular diseases, including stroke [11,12,13,14,15]. Non-enzymatic glycation and oxidation of proteins, nucleic acids, and lipids produce AGEs [16,17]. Receptors for AGEs or RAGEs, which are multiligand receptors of the immunoglobulin cell surface molecule superfamily, modulate atherosclerotic processes, trigger oxidative stress, inflammation, and apoptosis, and enhance tissue damage in focal cerebral ischemia [14,18].

Subsets of AGEs known as toxic AGEs (TAGEs) exert cytotoxic effects, and neurotoxicity has been reported in diabetic ischemic stroke although the exact molecular mechanisms are not known [18,19,20,21,22]. Interestingly, a link between the G protein-coupled receptor (GPCR) and RAGE in ischemic stroke pathology was revealed recently when RAGE was reported to modulate the signaling of a high-affinity GPCR for Leukotriene B_4_(LTB_4_), a potent proinflammatory lipid mediator [23,24,25]. RAGE has been proposed as a novel GPCR class of modulators and a target for future studies on therapeutic interventions. Intriguing organ crosstalk between the brain and kidney, renal dysfunction, and decreased renal clearance of AGEs cause their accumulation in circulation to occur in ischemic stroke [26,27]. A low glomerular filtration rate (GFR) in stroke was found to increase all-cause mortality and recurrent stroke risks [28]. This evidence indicates the interplay between AGEs, RAGE ligands, GPCR, and neurotoxicity in diabetic ischemic stroke [13,14,15,18,24,26].

This scoping review systematically maps the current and previous findings on AGEs, RAGE ligands, Leukotriene B_4_GPCR, organ crosstalk, and pathomechanisms in diabetic ischemic stroke to decipher the intricate interplay of molecular mechanisms. This will explicate the close interaction between diabetes mellitus and ischemic stroke as two complementary disorders associated with cerebrovascular complications. Experimental and clinical findings on several potentially useful therapeutic agents that target these mechanisms are suitably presented. The appraisal of key findings on all these components is required for the identification of key targets and to aid the development of effective comprehensive therapeutic strategies for improving patient outcomes in diabetic ischemic stroke.

## 2. AGEs, Toxic AGEs, and RAGEs

AGEs are irreversible covalent adducts formed endogenously through non-enzymatic glycation and glycoxidation of proteins, nucleic acids, and lipid biomolecules. Dietary AGE sources also exist, such as processed and barbequed foods [16,17,29]. AGEs such as carboxymethyllysine, pentosidine, and pyrraline, which do not exert direct cytotoxic effects, are considered non-toxic. They form a defense mechanism to trap aldehyde/carbonyl compounds with high reactivity and detoxify them [17,18,21]. The enhanced generation of AGEs in diabetes contributes to the development of micro- and macro-vascular complications, end-organ damage, and cardio-cerebrovascular diseases including stroke, as AGEs crosslink intracellular and extracellular matrix proteins, thus causing modifications in the structure, function, and mechanical properties of tissues [29,30]. Toxic AGEs (TAGEs), a subset of AGEs, exert toxic effects on certain cells and tissues through their interaction with receptors for AGEs (RAGEs) [18,19,20,21]. TAGEs, which bind to RAGEs, have been implicated in diabetes mellitus and its associated vascular complication pathogenesis [17,18,21]. Based on their effects and properties such as fluorescence, different types of AGEs exist, as shown in Table 1 [16,17,29,30].

For AGE measurements in human subjects, a noninvasive and effective tool termed skin autofluorescence (SAF) is available, which detects AGEs in the dermis layer [31,32]. SAF is a tool to establish the risk of chronic complications of diabetes mellitus for prompt intervention. Another investigation, namely, Blood AGE levels by ELISA, an enzyme-linked immunosorbent assay, employs the AGE antibodies [31,33]. Serum TAGE levels are also promising novel biomarkers to monitor the onset and progression of diseases related to lifestyle, measured using a specific antibody and competitive ELISA methods [17]. Tissue, plasma fluorescence, and SAF have been used to estimate AGE accumulation, and point-of-care testing (POCT) devices to monitor serum AGEs are currently available [31,32]. In both chronic cerebral infarction and silent brain infarction patients, higher SAF levels of AGEs have been reported. Significantly higher serum levels of fluorescent AGE pentosidine were found in acute ischemic stroke in comparison with healthy controls using the ELISA method [33].

Accumulated AGEs primarily interact with their receptors—RAGEs—to produce persistent vascular inflammation. Excessive downstream inflammatory indicators linked to the AGE–RAGE interaction have been linked to post-stroke inflammation, which is known to worsen ischemic brain damage and lead to cardiac injury [34]. By triggering signalling cascades through RAGE, AGEs mediate their detrimental effects. Numerous cell types in the peripheral and central nervous systems, including neurons, microglia, astrocytes, epithelial cells, mononuclear phagocytes, and endothelial cells, express RAGE, which is a 45 kDa immunoglobulin-type transmembrane receptor [35]. Due to the presence of multiple domains, isoforms, and polymorphisms, RAGEs bind to a discrete repertoire of ligands. The ligands for RAGEs also include many members of the S100/calgranulin family, oligomeric forms of A, high-mobility group box 1 (HMGB1), phosphatidylserine (PS), and lysophosphatidic acid. Interactions between AGEs and RAGEs result in diverse cell responses such as altered gene expression, migration, proliferation, and the activation of signaling pathways that cause oxidative stress, inflammation, apoptosis, atherosclerosis, and other vascular complications [35,36]. One of the numerous isoforms of RAGE is the soluble form of RAGE (sRAGE). Soluble RAGE also consists of several forms, including a spliced variant of RAGE, called endogenous secretory RAGE (esRAGE). These RAGE isoforms have extracellular domains but lack the intracytoplasmic and transmembrane domains. They can bind to ligands, including AGEs, and antagonize RAGE signalling. Scavenging receptors such as SR-AI are involved in the removal of AGEs. The pathophysiological roles of these RAGE isoforms in cardiovascular disease and the usefulness of soluble RAGE as disease biomarkers and therapeutic targets have all been indicated. ELISA techniques are used to assess the plasma levels of esRAGE and total sRAGE [34,35,36].

## 3. Diabetic Ischemic Stroke and RAGE-Mediated Ischemic Brain Damage

Stroke is a frequent, severe neurovascular disorder that is a major cause of disability and mortality. As per the TOAST classification, subtype 1 of ischemic stroke is associated with large-artery atherosclerosis [8] as follows: “Patients having clinical and brain imaging findings of either significant (>50%) stenosis or occlusion of a major brain artery or branch cortical artery, presumably due to atherosclerosis. Clinical findings include those of cerebral cortical impairment (aphasia, neglect, restricted motor involvement, etc.) or brain stem or cerebellar dysfunction. History of intermittent claudication, transient ischemic attacks (TIAs) in the same vascular territory, a carotid bruit, or diminished pulses helps support the clinical diagnosis. Cortical or cerebellar lesions and brain stem or subcortical hemispheric infarcts greater than 1.5cm in diameter on Computed Tomography(CT) or Magnetic Resonance Imaging (MRI) are considered to be of potential large-artery atherosclerotic origin. Supportive evidence by duplex imaging or arteriography of stenosis of greater than 50% of an appropriate intracranial or extracranial artery is needed. Diagnostic studies should exclude potential sources of cardiogenic embolism. Stroke diagnosis secondary to large-artery atherosclerosis cannot be made if duplex or arteriographic studies are normal or show only minimal changes“ [8]. Diabetes mellitus (DM) is associated with ischemic stroke subtype 1–large-artery atherosclerosis as it is the main cause of atherosclerosis. DM is also associated with subtype 3 of ischemic stroke—small vessel occlusion or lacunae—as the clinical diagnosis is supported by DM or hypertension history [8].

An increasing body of evidence shows the involvement of RAGE and its ligands in the pathogenesis of cardio-cerebrovascular, neurodegenerative, neuroinflammatory, and autoimmune disorders. sRAGE levels are elevated in the acute phase in the serum of stroke patients [37]. Neurons and glial cells in the brain express RAGE [38,39]. RAGE levels have been found to be elevated in biopsy samples taken from human patients who had a unilateral cerebral infarction as well as in the ischemic brain hemisphere of mice one day following an experimental stroke [40]. It is crucial to underscore the key roles of AGEs in oxidative stress, lipid oxidation, atherosclerotic processes, triggering of inflammation, and dysfunctions of mitochondria, as well as endothelial cells, in chronic cerebral ischemia. The interaction between AGE–RAGE and oxidative stress is also crucial when discussing the cardio-cerebrovascular consequences of diabetes. The carotid intima-media thickness (IMT) is employed as a measure of subclinical atherosclerosis since atherosclerosis is characterized by intramural thickening of the sub-intima. It is interesting to note that carotid IMT progression is a surrogate marker for risk stratification of both cerebrovascular and cardiovascular disease [41]. Following focal cerebral ischemia, the amount of irreversible damage and necrosis is increased by AGE-modified proteins and peptides [22]. In this respect, the number of skin AGEs measured via skin autofluorescence (SAF) is a helpful metric in the treatment of acute stroke as a predictor of the outcome of ischemic stroke in patients with diabetes mellitus [34].

Numerous variables in diabetes patients, such as hyperglycemia, vascular risk factors such as hypertension and dyslipidemia, and genetic, demographic, and lifestyle variables all play a role in the overall cardio-cerebrovascular risk to varying degrees [42,43,44]. The importance of secondary prevention methods against debilitating strokes in diabetic individuals is highlighted by the fact that their risk of ischemic stroke is roughly double that of people without diabetes [10]. The relationship between diabetes and ischemic stroke is reciprocal, and diabetes may cause more subtle brain damage indicated by lacunar infarcts, which raises the risk of dementia and causes a faster loss of cognitive function [9]. In an ischemic stroke, the necrotic core is surrounded by an inflammatory zone. The initial injury is aggravated by delayed cell death. As a mediator of ischemic brain damage, a sensor of necrotic cell death, and a contributor to both inflammation and ischemic brain damage, RAGE plays key roles. The high-mobility group box 1 (HMGB1) ligand released from necrotic cells seems to be the most obvious RAGE activator in stroke. In experimental models of cerebral ischemia, RAGE was shown to be a key mediator. An independent predictor of poor outcomes for stroke in type 2 diabetes is an elevated blood RAGE level [45]. After a stroke, RAGE may play a role in a more vigorous inflammatory response that is mediated by HMGB1 [46].

HMGB1, a non-histone nuclear protein, is a crucial proinflammatory alarmin in cerebrovascular ischemic disorders [43]. RAGE and the toll-like receptors TLR-2 and-4 are the main HMGB1 receptors investigated in brain injury. These receptors are ubiquitously expressed on microglia, astrocytes, and neurons in the CNS [47,48]. In the early stages of stroke, several neurons undergo sustained oxidative toxicity and hypoxia. HMGB1 modifications occur via acetylation and phosphorylation, which decrease its affinity for DNA when microglia and astrocytes are activated. As neuron cell membranes are destroyed, HMGB1 loosely bound to chromosomes is passively released into the extracellular space within 2–4 h after ischemia–reperfusion [46]. Hyperglycemia causes early extracellular HMGB1 secretion from ischemic brain tissue and also increases the extracellular accumulation of glutamate, an excitatory amino acid that plays a central role in neuronal death. This increases infarct volume, neurological deficits, cerebral edema, and blood–brain barrier (BBB) disruption, which is a critical event in the growth of cerebral edema in the early stages of ischemic brain injury [46,49]

The following molecular processes occur after HMGB1 release during ischemic events [46]:HMGB1 interacts with the receptors TLR-2 and TLR-4, which are expressed on monocytes through the adaptor protein myeloid differentiation factor 88 (MyD88) and elevates serum levels of TNF-α, IL-6, and IL-1β l, which leads to cerebral vessel occlusion.TNF-α and IL-1β, which are HMGB1-induced cytokines, can indirectly promote the upregulation of matrix metalloproteinase MMP-9. MMP upregulation causes a rise in infarct size, brain edema, and recombinant tissue plasminogen activator-induced bleeding, which hastens damage to the tight junction protein Occludin and other neurovascular substrates.HMGB1 stimulation of TNF-α, IL-1, IL-6, and IL-8 production induces the expression of inducible NOS (iNOS) during ischemic brain damage. The induction of iNOS and TNF-α occurs mainly in microglia. This produces an inflammatory response and BBB disruption, leading to brain infarction aggravation.RAGE expression, which is low in cells under physiological conditions, rises in response to an increase in HMGB1 ligand molecules, for which RAGE has a strong affinity. HMGB1 binding to upregulated RAGE leads to the activation of several signal-transduction pathways including Mitogen-activated protein kinases (MAPKs), phosphatidylinositol 3 kinase/protein kinase B (PI3K/Akt)-p38 kinase, SAPK/JNK, extracellular regulated protein kinases1/2 (ERK 1/2), Akt, Ras-related C3 botulinum toxin substrate (Rac), Cell division cycle 42 (Cdc42), and Just another kinase/signal transducer and activator of transcription 1 (JAK/STAT1)-mediated signal transduction pathways. Finally, these processes result in the translocation of nuclear factor kappa-light-chain-enhancer (NF-κB), which triggers the expression of inflammatory cytokines and chemokines that help immune cells mature, migrate, and express surface receptors, and cause neuritis [50].

RAGE has been linked to neuronal cell death following ischemia and is expressed by endothelial and microglial cells in the human brain. Endothelial RAGE performs two distinct functions in ischemia: It transports endogenous secretory RAGE (esRAGE), a neuroprotectant, to the brain and also induces vascular injury and neuronal damage [37]. The interaction between toxic AGEs (TAGE) and RAGE elicits vascular inflammatory processes and oxidative stress in many cell types, including endothelial, causing microvascular circulation derangement [18]. Previous studies in mice models of distal permanent middle cerebral artery blockage, which generate focal ischemic stroke, showed a critical role of neuronal and microglial RAGE in ischemia-induced neuronal death and inflammation [51]. In a mouse model of cerebral ischemia, HMGB1 was found to be released from ischemic brain tissue and increased in the serum of stroke victims. Ischemic brain damage was lessened by a neutralizing anti-HMGB1 antibody and HMGB1 box A, an antagonist of HMGB1 at the receptor RAGE. Decoy receptor soluble RAGE and genetic RAGE deficiency caused a reduction in infarct size [52].In vitro expression of RAGE mediated the toxic effect of HMGB1 on microglial cells. Brain macrophages mediate the HMGB1–RAGE-induced injury. Data from in vivo animal model experiments using chimeric mice strongly indicated that for the development of ischemic brain damage, RAGE expressed by immigrant macrophages is required, possibly for binding the HMGB1 released from necrotic cells in the ischemia core [52]. HMGB1–RAGE signaling in stroke links macrophage activation with necrosis, providing a therapeutic target. The HMGB1 antagonist box A, a neutralizing anti-HMGB1 antibody or soluble RAGE, may be a specific tool to target inflammation and delayed cell death [52].

## 4. AGE-RAGE System, Blood–Brain Barrier Dysfunction, Neuroinflammation and Neurodegeneration in Ischemic Stroke

Ischemic stroke contributes to the disintegration of the neurovascular unit, which consists of endothelial cells of the blood–brain barrier (BBB), neurons, astrocytes, myocytes, pericytes, and extracellular matrix components. BBB, a highly selective semipermeable border of endothelial cells, is the blood–brain interface, mediating communication between central and peripheral nervous systems. Synaptic-neuronal dysfunction, damage, and loss are caused by a vicious cycle of degeneration driven by an irreversible BBB breakdown, chemical imbalance in the internal milieu of neurons, and neurovascular inflammatory immunological responses [36,53]. The molecular mechanisms involved in the neurodegenerative processes of ischemic stroke that have been studied are explained as follows:During thrombosis, amyloid beta (A*β*) peptides produced in blood vessels are discharged into the brain and momentarily accumulate there. During an ischemic stroke, they act as additional damaging factors by forming ion channels. RAGE is implicated in the neurotoxic immunoinflammatory cascades and in the amyloidogenic pathway. RAGE is the primary influx transporter for A*β* across the BBB. RAGE-soluble A*β* binding mediates the pathophysiologic cellular responses. The RAGE–A*β* interaction causes oxidative damage to RAGE-expressing neurons and activates microglia, which both directly and indirectly contributes to neuron death. RAGE inhibitors can prevent the production of cytokines and chemokines, oxidative stress, and A*β* BBB transport by blocking the pathophysiological effects of the RAGE–A*β* interaction in the afflicted vasculature [36,53].RAGE is connected to both independent neurotoxic immunoinflammatory cascades and the amyloidogenic pathway in neurodegenerative diseases [48]. RAGE is overexpressed in neurons, microglia, astrocytes, and the BBB vasculature when endogenous ligands such as AGE, S100, or A*β* bind to physiologically expressed RAGE [36,48]. RAGE is more prevalent at the BBB, which causes an influx of monocytes and A*β* into the brain. However, RAGE is more active in neurons, where it enhances A*β*-producing β-secretase enzyme (BACE1) activity, tau hyperphosphorylation, and neuroinflammation and impairs neuronal function. Ischemia-induced A*β*/tau pathology, similar to Alzheimer’s disease, is reported to be involved in post-stroke cognitive impairment [36].Hyperglycemia-induced overexpression of mitochondrial superoxide in endothelial cells causes microvascular injury in diabetes mellitus. Superoxide overproduction can activate AGEs formation and protein kinase C (PKC) signaling. PKC induces BBB damage through the disruption of tight junction (TJ) proteins, phosphorylation of cytoplasmic adaptor zona occuldens-1 (ZO1), and enhanced expression of vascular endothelial growth factor. Upregulated and activated RAGEs induce oxidative stress and activate the NF-κB pathway. TNF-α, IL-6, and IL-1 transcription are increased when the NF-κB pathway is activated in vascular cells [18,36].AGEs cause neurodegeneration by pathways unrelated to RAGE, such as protein modification and cross-linking, contributing to toxic effects of A*β*, affecting cellular damage, tissue stiffness, vascular pathological processes, and the formation of aggregates [54,55,56].The AGE–RAGE system may also participate in Apolipoprotein E-ε4 (APOE-ε4 allele)-associated pathological processes of dementia. Associations between higher skin autofluorescence due to AGEs, lower cognitive function, and APOE-ε4 status have been reported [57]. APOE-ε4 impacts the risk of dementia associated with stroke in patients, and both pre- and post-event dementia were found to be linked to APOE-ε4 homozygosity. The relationships were not related to the burden of the cerebrovascular system and may be explained by increased neurodegenerative disease or damage susceptibility [58].

## 5. Interplay of Leukotriene B_4_ Receptor 1 (BLT1) and RAGE in Ischemic Stroke

The potent, short-lived, pro-inflammatory lipid mediators Leukotrienes (LTs) are expressed in macrophages, neutrophils, and mast cells. LTs have a part to play in cerebrovascular disorders, as well as innate immune responses. They are reported to have roles in the pathogenesis of ischemic stroke and other atherosclerosis processes in the cardio-cerebral vasculature [23,24]. Leukotriene B4 (LTB_4_) is a group of LTs with the specific receptor BLT1 [23]. Early and sustained increases in LTB4 levels have been associated with poor clinical outcomes in ischemic stroke patients [25,59].

Leukotriene B_4_ receptor 1 (BLT1), a G protein-coupled receptor (GPCR) with a high affinity for LTB_4_,has important roles in inflammatory and immune reactions. The LTB_4_–BLT1 axis was found to enhance inflammation, but the binding proteins that modulate LTB_4_–BLT1 signalling were unidentified [23,24]. Recently, RAGE was shown to be a BLT1-binding protein that regulated BLT1 signalling. RAGE acts as a molecular switch for BLT1 by inhibiting BLT1-dependent activation of NF-κB and stimulating BLT1-dependent chemotaxis. RAGE was identified as a new class of GPCR modulator because of its capacity to bind to GPCRs other than BLT1 [23,24]. Due to these recent findings on leukotriene receptors, they are currently viewed as new drug targets [59,60].

## 6. Neurotoxicity of AGEs Demonstrated by Animal and Human Models

The proper functioning of both glial cells and neurons in the grey matter of the brain is of paramount importance in preventing neurodegenerative diseases. AGEs were reported to have neurotoxicity-potentiating effects causing brain damage during diabetic ischemic stroke in an animal study [22]. The maximal neurotoxicity window was within a few hours after AGE exposure with downregulation. Aminoguanidine (AG), a dicarbonyl scavenger and AGE crosslink breaker/inhibitor, attenuated the infarct volume in AGE-treated animals [22]. Among the TAGEs, AA-AGE and AGE-2 appeared to be particularly neurotoxic [20]. In cortical neuronal cells, AGE-2 exhibited neurotoxic effects, which were more potent than that of CML and Glu-AGEs. The neurotoxic effects of serum AGE fractions were attenuated entirely by anti-GA-AGE antibody addition but not by other types of AGE antibodies in another line of evidence wherein GA-AGE involvement in neurodegeneration was shown [20,22]. Assessment of the effects of AGE-2 and AGE-3 on cultured Schwann cells in the white matter of the brain showed that the production and proliferation of proinflammatory cytokines and cell viability were significantly affected by TAGE treatments [20,61].

The mechanisms for AGE neurotoxicity are possibly related to nitric oxide and polyamine metabolisms [22]. Excessive production of nitric oxide (NO) by brain parenchyma is a characteristic of ischemic stroke [62,63,64]. Endothelium-derived NO bioavailability and activity are decreased by AGEs. This effect of AGEs on NO may be significant to atherogenesis since NO inhibits several of the processes that lead to atherosclerosis, including leukocyte adherence to the artery wall, vascular smooth muscle development, and platelet adhesion-aggregation. The antiproliferative actions of NO are inhibited by matrix-bound and soluble AGEs [62,63,64]. Impaired vasodilation in diabetes may be a result of the reduction in NO activity by AGEs [65].By speeding up mRNA degradation and decreasing endothelial NO synthase (eNOS) activity, AGEs shorten the half-life of eNOS mRNA. Through the binding of CML to endothelial RAGE, AGEs hinder NO generation by reducing the phosphorylation of serine residues in eNOS, which deactivates the enzyme [62,63,64]. AGEs may also quench and inactivate NO produced by the endothelium [65]. The flavanoid Quercetin greatly inhibited total AGEs and high-molecular-weight AGE in vitro [66]. The highly favorable therapeutic potential of quercetin and its analogues against NO synthase from natural sources for ischemic stroke treatment has been reported recently using molecular docking analysis scores involving ligand–protein interactions [67].

There is a lack of clarity on in vivo mechanisms directly linking AGEs to polyamine metabolism in diabetic ischemic stroke. However, in diabetes mellitus, in vitro formation of the AGE Pyralline was inhibited by polyamines–spermine, spermidine, and L-Arginine, the biological precursor of NO [64]. Polyamine oxidation is enhanced in cerebral ischemia and the production of their toxic metabolites may enhance neuronal death [68]. Therefore, the AGEs polyamine and NO are possibly linked by unknown mechanisms. Polyamine mechanisms have been described as possible pathomechanisms in diabetic ischemic stroke. The polyamine process in eukaryotes produces spermidine, which triggers the hypusination of eukaryotic initiation factor 5A (eIF5A), a translation factor. Inhibition of the eIF5A hypusination pathway is a new pharmacological target for stroke therapy [69]. The polyamine inhibitor was also reported to ameliorate brain infarction size in a mouse model [70].

The multiple molecular mechanisms involving the AGE–RAGE axis and other pathomechanisms involved in diabetic ischemic stroke are represented in Figure 1.

## 7. Brain–Kidney Organ Crosstalk, Renal Dysfunction, and Plasma AGEs

The brain and kidneys are inextricably linked through complex interactions and shared risk factors such as diabetes and hypertension [26,27]. Knowing how the brain and kidney communicate bidirectionally after cerebral injury will have a practical impact on therapeutic strategies for improving stroke patient outcomes. Post-stroke brain–kidney crosstalk has been suggested in both experimental and observational studies. After a stroke, the roles of the central autonomic network, autoregulation, inflammatory and immunological responses, extracellular vesicles, and their cargo microRNA have been studied [26]. Yet, current evidence is mainly associative, and evidence on mechanisms underlying direct brain–kidney interaction is limited. Patients with type 2 diabetes who also had early chronic kidney disease (CKD) in deteriorating stages had a greater risk of incident stroke [71]. Renal dysfunction influenced stroke prognosis as a low estimated glomerular filtration rate (eGFR) <45 mL/min/1.73 m^2^ in stroke patients was associated with increased all-cause mortality risk and recurrent stroke [28]. In patients with decreased renal function, fluorescent AGE levels were reportedly elevated due to reduced clearance and increased carbonyl stress. Fluorescent AGE levels were inversely related to eGFR in those studies [72,73,74]. Interestingly, one of those studies also reported the presence of a new class of AGEs called Melibiose-derived glycation products (MAGEs, derived from melibiose, a disaccharide of glucose and galactose), which were significantly higher in patients with hypertension [74]. The quality of dialysis water and infusion fluid during renal replacement therapy modalities influences plasma AGE levels in end-stage renal disease patients. The highest AGE fluorescence and CML levels were found in hemodialysis patients treated with standard dialysis fluid rather than those with ultrapure fluid and hemofiltration [75].

## 8. Therapeutic Agents and Their Effects against the AGE–RAGE Axis and Other Key Targets in Diabetic Cardio-Cerebrovascular Complications

Several therapeutic agents/drugs, including several investigational/candidate therapeutic agents targeting the crucial molecular mechanisms, have been reported to be effective against diabetic cardio-cerebrovascular complications including ischemic stroke [76,77,78]. They are elaborated on in Table 2, Table 3, Table 4, Table 5 and Table 6. Some of the agents can be obtained from natural sources or diets. Several vitamins, including antioxidant vitamins E and C, which can reduce oxidative stress, have also been studied for their possible antiglycation effects and impact on AGE levels, as detailed in Table 4.

Metformin therapy in type 2 diabetes mellitus patients was found to be associated with lower risk of acute ischemic stroke during COVID-19 in a retrospective cohort study [100]. Iminosugars and other sugar derivatives form an additional avenue for therapeutic agent development, which involves the precise modification of carbohydrate structures and the modulation of glycosidase activity. They offer broad therapeutic applications including anti-diabetic and immunosuppressive activities. Iminosugar analogs, which have nitrogen atoms substituted for oxygen atoms in the sugar ring structures, are potent inhibitors of the enzyme glucosidase that catalyzes the hydrolysis of glycosidic bonds and glycosyltransferases that catalyze glycosidic bond formation using sugar donors, in vivo [101,102,103,104,105].

Although there is a host of potential therapeutic agents against diabetes–ischemic stroke conditions, clinical evidence of most of the agents listed in Table 2, Table 3, Table 4, Table 5 and Table 6 is limited. In terms of clinical efficacy and practicality, mixed results were yielded in some of the conducted trials. The limitation of aminoguanidine for clinical usage was that it interfered with important regulatory systems and yielded some toxic side effects. The small size and short follow-up periods are common limitations of most of the clinical trials conducted. Yet, AGE–RAGE antagonists remain interesting therapeutic options for diabetes–stroke-associated complications [76,77,78]. An important factor that remains for consideration in stroke patients is better management of their motor impairments. Therefore, in addition to the employment of suitable molecular therapeutic targets, improved protocols for motoring recovery becomes an important component in the therapeutic strategies for stroke patients.

## 9. Myokines, Muscle-Organ Crosstalk, Neuromuscular Electrical Stimulation, and Improved Motor Recovery in Stroke Patients

Although the underlying mechanisms are not fully understood, electrical stimulation techniques have demonstrated good therapeutic potential in post-stroke motor rehabilitation. A recently developed promising method for restoring upper limb control and embodiment in stroke patients with long-term deficits combines a carefully calibrated, innovative neuromuscular electrical stimulation (NMES) technology with conventional rehabilitation techniques [106]. NMES induces circulating myokine secretion by repeated skeletal muscle contraction [107]. These recent advances are indicative of the emerging roles of myokines and their therapeutic potential. Evidence shows myokine production from skeletal muscle in response to exercise or neuromuscular electrical stimulation, which allows for crosstalk between the muscle and other organs, including the brain.

Myokines, which are cytokines or other small proteins and peptides produced and released by skeletal muscle, exert autocrine, paracrine, or endocrine effects [108]. The myokines IL-6, Brain-derived neurotrophic factor (BDNF), and Irisin modulate vital processes such as brain neuroplasticity, hypertrophy, angiogenesis, and inflammatory and extracellular matrix regulation with exercise-induced myokines having therapeutic potential for muscle wasting [109]. The potential mechanism for Irisin’s therapeutic effect on vascular complications of diabetes has been investigated. Irisin significantly alleviated inflammasome signaling, AGE-induced oxidative stress, endothelial dysfunction, and increased eNOS and NO production in a dose-dependent manner, thus demonstrating the Irisin–AGE interplay [110]. By increasing levels of BDNF, which guards nerve cells, Irisin can reduce brain damage in ischemic stroke [111]. Low serum Irisin levels were found to be a predictor of poor early functional outcome in ischemic stroke patients [112] adding value to the potential of its development as a prognostic biomarker, as was Chemokine (C-C motif) ligand 2 (CCL2) levels in blood or cerebrospinal fluid, which were found to correlate with the clinical symptoms of stroke patients [113]. Finding new myokines, their specific receptors, and their neuroprotective functions may reveal novel therapeutic targets and biomarkers and open up new avenues for clinical interventions.

## 10. Plausible Comprehensive Therapeutic Strategies for Improved Management of Diabetic Ischemic Stroke

Diabetes, a well-established risk factor for ischemic stroke, has also been associated with the worsening of short- and long-term outcomes after the stroke event, making diabetes–ischemic stroke a deadly combination. Risk of cardiovascular or non-cardiovascular events, stroke recurrence, and death were at higher rates in patients with diabetes as compared to those without in a large study [114]. Finding an effective recipe of strategies for diabetic ischemic stroke patient management is, therefore, pivotal to combating the associated morbidity–mortality. The extensiveness of molecular mechanisms and the diversity of associated factors and their interplay in diabetic ischemic stroke thathave been described so far plausibly necessitate comprehensiveness in therapeutic strategies to improve patient outcomes. The key underlying target processes to focus on in diabetic ischemic stroke management can be categorized as neuroinflammation, oxidative stress, and improved neuromuscular stimulation for rehabilitation. An improved system for continual blood-level monitoring of promising biomarkers and the administration of inhibitor drugs/nutrients to target key axes are the other components required. Clinical laboratories will need to validate the novel biomarker testing in the current scenario to enable improved monitoring of the patients. Five key therapeutic/neuroprotective components for effective diabetic ischemic stroke management are identified and shown in Figure 2.

These five components show potential for application in future therapeutic interventions of diabetic ischemic stroke after the conduction of suitable large-sized clinical trials with longer follow-up periods and appropriate validation:Rapid screening of patients for serum AGEs, TAGEs, and risk stratification with point-of-care testing (POCT) devices validated for routine clinical use for monitoring disease progression and treatment effectiveness, as well as the employment of HMGB1 and CCL2 blood levels as prognostic markers for stroke patients.Administration of inhibitor drugs and nutrients against RAGE–ligand axes, NO synthesis, and polyamine oxidation as adjunctive therapy along with primary therapy.Improved management of renal dysfunction, the use of appropriate dialysate fluid to clear plasma AGEs more efficiently, and the formulation of renal dialysis modalities to improve clearance of low-molecular-weight fluorescent AGEsAnti-inflammatory therapy with agents from natural sources such as the plant flavanoid Quercetin or its analogues, which can target HMGB1–RAGE, LTB_4_–BLT1, and NO synthesis for selective neutralization of pathogenic immune signaling, tissue preservation, and neurological recovery. In addition, antioxidant therapy, which can combat not only the oxidative stress associated with hyperglycemia but also stress due to polyamine oxidation and NO synthesis, offering a key node in the preventive and therapeutic strategies for diabetes-related fatal cardio-cerebrovascular events.Neuromuscular stimulation interventions in diabetic ischemic stroke patients with possible employment of myokine Irisin as a biomarker for monitoring the impact of exercise type and amount.

## 11. Conclusions

Critical reappraisal of the myriad of molecular cascade events involving AGEs and RAGE-ligands reveals the intricate interplay in diabetes–ischemic stroke pathogenesis and resultant poor disease outcomes. AGEs, RAGE–ligand interactions, nitric oxide, and polyamines are pivotal molecular targets in the explication of associated key pathomechanisms. Inflammatory proliferative processes, oxidative damage, and cognitive decline can all be attributed to derangement in these key molecular mechanisms. An effective therapeutic strategy for diabetic ischemic stroke needs comprehensiveness to combat neurodegenerative damage and enhance motor recovery to improve the quality of outcomes in diabetic ischemic stroke patients. Agents to mitigate the AGE–RAGE axis, improved biomarkers for risk stratification, better renal dysfunction management, adjunctive anti-inflammatory–antioxidant therapies, and innovative neuromuscular stimulation for rehabilitation therefore become essential components to feature in the comprehensive therapeutic strategy for diabetic ischemic stroke.

## Figures and Tables

**Figure 1 biomolecules-12-01712-f001:**
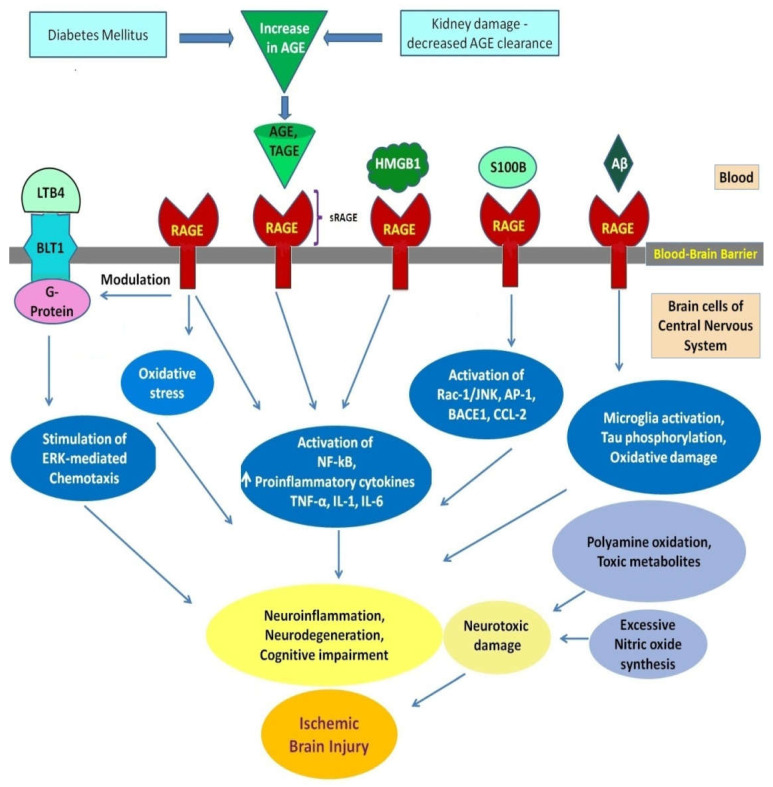
AGE–RAGE, G-protein coupled Leukotriene B4 receptor BLT-mediated molecular and pathological mechanisms in neurons, astrocytes, microglia, endothelial cells, and blood–brain barrier of central nervous system in diabetic ischemic stroke: (1) Modulation of G protein-coupled Leukotriene B4 (LTB4) receptor BLT1 by RAGE. (2) Binding of AGE to RAGE. sRAGE, Soluble RAGE—the extracellular/cytosolic domain of RAGE. (3) Binding of RAGE with ligands: AGE/TAGE, Advanced Glycation End Product/Toxic Advanced Glycation End Product; extracellular HMGB1, High-Mobility Group Box 1 protein released from necrotic cells; S100B protein from S100/Calgranulin family of proteins; A*β,* Amyloid beta *peptides*. (4) Molecular, pathological mechanisms involving ERK, the Extracellular-Signal-Regulated Kinase; NF-κB, Nuclear factor kappa-light-chain-enhancer; TNF-α, Tumor Necrosis Factor-α; IL-1, IL- 6, Interleukins 1,6; Rac-1/JNK, Rac-1 G protein/Jun N-terminal kinase; BACE1, Beta-site APP Cleaving Enzyme 1; AP-1, Activation Protein 1 and CCL2, Chemokine (C-C motif) ligand 2.

**Figure 2 biomolecules-12-01712-f002:**
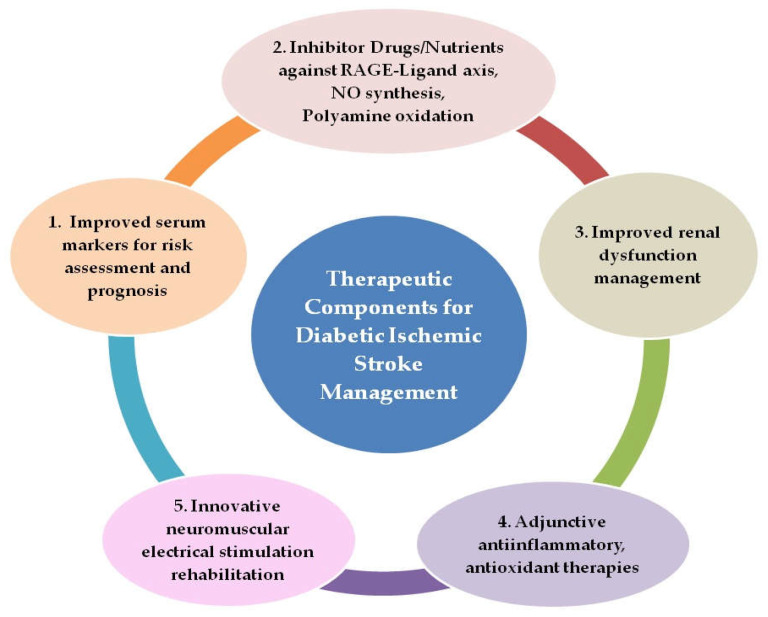
Components for comprehensive outcome-improvement therapeutic strategy in diabetic ischemic stroke patients.

**Table 1 biomolecules-12-01712-t001:** AGEs and Toxic AGEs: Types and examples.

AGEs	
Cross-linking	Non-cross-linking
*Fluorescent:* Vesperlysine, Pentosidine, Crossline	*Non-fluorescent:* N-fructosyl-lysine (FL) N carboxyethyl-lysine (CEL) N-carboxymethyllysine (CML) PyrralineImidazolone
*Non-fluorescent:* Imadazolium dilysine crosslinks Alkyl formyl glycosyl pyrroles Arginine-lysineimidazole crosslinks	
**Toxic AGEs** Glyceraldehyde-derived AGE-2 Glycolaldehyde-derived AGE-3 Acetaldehyde-derived AA-AGE	

**Table 2 biomolecules-12-01712-t002:** Experimental findings on investigational/candidate therapeutic agents against AGE–RAGE axis in diabetic cardio-cerebrovascular complications.

Investigational Therapeutic Agent, *Category*	Model	Result	References
Aminoguanidine, *Dicarbonyl scavenger* *AGE cross link breaker*	Rat model of focal cerebral ischemia	Aminoguanidine attenuated infarct volume in AGE-treated animals in dose- and time-related manner with cerebral blood flow.	[22]
ALT-711 (Alagebrium), *AGE cross-link breaker* and Aminoguanidine	Mouse, Streptozotocin- induced diabetic apolipoprotein E–deficient (apoE-/-)	ALT-711 and Aminoguanidine reduced vascular AGEs-CML, CEL accumulation, skin collagen solubility and attenuated atherosclerosis.	[79]
Soluble RAGE, *Competitive inhibitor* *of RAGE*	Cultured human umbilical vein endothelial cells, (HUVECs) and mouse model of partial carotid artery ligation	sRAGE significantly inhibited oscillatory shear stress (OSS)-induced expression of RAGE and HMGB1 in HUVECs. RAGE expression was markedly elevated in the vicinity of atherosclerotic plaque, and administration of sRAGE inhibited plaque development.	[80]
endogenous secretory RAGE (esRAGE), *Decoy splicing* *variant RAGE*	RAGE knockout, wild-type and human esRAGE overexpressing transgenic, mice	In the BBB system, esRAGE transfer from vascular to the brain side was shown to be RAGE-dependent. esRAGE served as a decoy to prevent neuronal cell death induced by ischemia.	[81]

**Table 3 biomolecules-12-01712-t003:** Experimental and clinical findings on therapeutic drugs against AGE–RAGE axis in diabetic cardio-cerebrovascular complications.

Therapeutic Agent, *Category*	Model	Result	References
Metformin, *Hypoglycemic drug* Sulphonylurea, *Hypoglycemic drug* Acetylsalicylic acid/Aspirin, *Anti-inflammatory drug* Acarbose, *AGE inhibitor with* *chelating properties,* *inhibitor ofα-glucosidase* Clopidogrel, *Glycation-preventing inhibitor* *of excessive platelet aggregation*	in vitro, serum of DM patients with complications, ischemic stroke Single-arm T2DM, hypertension	SR-AI scavenging receptor concentration was significantly reduced in metformin-treated diabetic patients. Correlation was found between ischemic stroke and Melibiose-derived glycation product (MAGE) content. Significant association was found between sulphonylurea intake and a higher sRAGE concentration due to counteraction of effects of AGE formation by sulphonylurea. Aspirin use was significantly associated with decreased total AGE fluorescence which confirmed its effectiveness on glycation inhibition. Significant correlation was found between acarbose intake and fluorescence of total soluble AGEs or soluble pentosidine. Clopidogrel lowered protein-bound AGE and protein-bound Pentosidine fluorescence due to its action of inhibiting fluorescent AGE formation.	[74]
*Hypoglycemic drugs:* Pioglitazone/Thiazolidinediones (TZD) and Metformin Dipeptidyl peptidase 4 (DPP4) inhibitor/ Alogliptin	RCT, T2 DM Single-arm, T2DM	Pioglitazone proved to be superior in amelioration of oxidative stress in comparison with no medication. Pioglitazone and metformin were equally effective in advanced oxidation protein products (AOPP) and AGE decrements. sRAGE concentrations decreased with alogliptin treatment and was associated with HbAlc concentration changes. Albuminuria was reduced after the treatment.	[82,83]
Atorvastatin, *Lipid-lowering drug*	RCT, T2DM, hypercholesterolemia	Atorvastatin significantly reduced AGE, total cholesterol, LDL and triglycerides levels.	[84]
Epalrestat, *Aldose reductase inhibitor*	Observational, diabetic peripheral neuropathy (DPN)	Epalrestat decreased CML and slowed the progression of peripheral diabetic neuropathy.	[85]

**Table 4 biomolecules-12-01712-t004:** Experimental and clinical findings on vitamins as therapeutic agents against AGEs in diabetic patients with and without renal complications.

Vitamin	Model	Result	References
Benfotiamine *Lipid-soluble precursor of Thiamine*	RCT T2DM	Benfotiamine group had significantly decreased CML-AGE levels and placebo group had significantly decreased sRAGE levels.	[86]
Pyridoxamine, *a broad inhibitor of* *advanced glycation*	RCT DM, overt nephropathy	Pyridoxamine reduced the serum AGEs-CML and CEL in addition to TGF-β1and urine creatinine levels. There were no differences in urinary albumin excretion.	[87]
α-lipoic acid plus Pyridoxine	RCT Diabetic nephropathy	The AGEs-Pentosidine and CML were significantly decreased in the supplemented group.Urinary albumin, serum malondialdehyde (MDA), and systolic blood pressure significantly decreased in the supplemented group compared to the placebo group. Serum NO was increased in the supplemented group compared to the placebo group.	[88]
Vitamin E	Invitro, Diabetic nephropathy patients	AGE-Bovine serum albumin exposure enhanced cellular secretion of the renal tubular injury markers Hepatitis A virus cellular receptor 1 (HAVCR1) and Lipocalin 2(LCN2) which was significantly reduced by vitamin E treatment.	[89]
Ascorbic acid/ Vitamin C	Plasma levels, T2DM	Antiglycation role of ascorbic acid was evident as increasing the ascorbic acid concentrations greatly diminished protein glycations and inhibited AGE in dose-dependent manner.	[90]

**Table 5 biomolecules-12-01712-t005:** Experimental findings on natural compound-based AGE–RAGE inhibitors against diabetic, cardio-, and cerebrovascular complications.

Natural Compound	Model	Result	References
Gallic acid, *Polyphenol*	Cell culture H9C2 (2-1), heart	Significant attenuated expressions of RAGE and other cytokines were found in Gallic acid pre-treated cells.	[91]
*Cucurbita,* *Pumpkin Polysaccharides (PPs)*	in vitro	Inhibitory effects of PPs on AGEs formation were higher and stronger than the positive control, Aminoguanidine.	[92]
*Terpenoids:* Ursolic acid Oleanolic acid (OA)	Rat, in vitro and in vivo in vitro	Urosolic acid showed the most potent Aldose reductase inhibitory action and suppressed RAGE expression. OA almost completely inhibited AGE formation.	[93,94]
Berberine, *Alkaloid*	Rat model of diabetes	Berberine inhibited AGE accumulation, and improved antioxidant capacity with protective effects against diabeticrenal damage.	[95]
Carnosine, *Pleiotropic dipeptide*	Rat, ex vivobrain homogenates and primary neuronal/astrocytic cultures	Carnosine treatment exhibited significant cerebroprotection against histological and functional damage in both permanent and transient ischemic models.	[96]
Cyanidin-3-O-glucoside *Anthocyanin of red rice*	in silico analysis	Cyanidin-3-O-glucoside could bind to RAGE at the same residue as AGEs -Argypirimidine and Pyrralline, which indicated that it might have a biological function as a competitive inhibitor of AGEs-RAGE interactions through AGEs-cyanidin-3-O-glucoside-RAGE complex establishment.	[97]

**Table 6 biomolecules-12-01712-t006:** Experimental findings on investigational/candidate therapeutic agents against targets other than AGE–RAGE Axis in ischemic stroke.

Investigational Therapeutic Agents, *Category*	Model	Result	Ref.
*HMGB1 axis antagonists:* Soluble RAGE (sRAGE); recombinant A neutralising anti-HMGB1 antibody	Ischemic stroke (IS) patients and C57BL/6J mouse model of focal ischemic stroke Mouse model of cerebral ischemia	Within 48 h of IS, sRAGE and HMGB1 plasma levels considerably rose. Recombinant sRAGE significantly reduced immune cell infiltration, enhanced mouse injury outcome, and ameliorated the negative effects of recombinant HMGB1. Ischemic brain damage was ameliorated by a neutralising anti-HMGB1 antibody and HMGB1 box A, an antagonist of HMGB1 at RAGE. Infarct size was decreased by soluble RAGE and genetic RAGE deficiency.	[37,52]
*BLT Receptor antagonist* LY255283	Ischemic stroke patients, Rat stroke model	Compared to early post-stroke levels, plasma LTB4 levels surged quickly, roughly doubling in just 24 h. LY255283 reduced the size of the infarct.	[59]
*Polyamine-eIF5A-hypusine Axis inhibitors:* N1-guanyl-1,7-diaminoheptane (GC7) *N*1-Nonyl-1,4-diaminobutane (C9-4) N-2-mercaptopropionyl glycine (N-2-MPG)	in vivo transient focal cerebral ischemia mouse model Photochemically induced thrombosis model mice in vitroscreening of clinically approved drugs in rat model of middle cerebral artery occlusion, neuronal cell line (HTB11), glial cell line (HTB14)	A single GC7 pre- or post-treatment significantly reduced infarct volume post-stroke. Post-treatment GC7 significantly improved motor and cognitive post-stroke deficits. C9-4 reduced the volume of brain infarction significantly, demonstrated to be potent inhibitor of polyamine-oxidizing enzymes. in vitro, N-2-MPG significantly inhibited cytotoxicity of 3-aminopropanal (reactive catabolite ofpolyamines) and in rats it reduced infarct volume, even when the agent was administered after ischemia onset.	[69,70,98]
*Nitric oxide* *synthase antagonists:* Aminoguanidine (AG) Quercetin and its analogues	Rat model cerebral ischemia Validated Homology modeling for human inducible NO synthase (NOS), Molecular docking studies	NO production via inducible NOS was suppressed by Aminoguanidine. NO contributed to delay in recovery from brain neuronal damage in hippocampus following ischemic brain injury. Highly favourable interactions of Quercetin and its derivatives occurred on NOS involving ligand–protein and docking scores. Quercetine and derivatives were found to be suitable molecules for testing as anti-stroke agents.	[67,99]

## Data Availability

Not applicable.

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
