# Peer review of "Receptor for Advanced Glycation End Product, Organ Crosstalk, and Pathomechanism Targets for Comprehensive Molecular Therapeutics in Diabetic Ischemic Stroke"

_biomolecules, 2022, doi:10.3390/biom12111712_

Round 1
Reviewer 1 Report (New Reviewer)
The submitted review entitled ‘‘Receptor for Advanced Glycation End Product, Organ Crosstalk and Pathomechanism Targets for Comprehensive Molecular Therapeutics in Diabetic Ischemic Stroke’’ by Saiqa R Shah et al. examined, comments to the review are given below.
The authors have described the risk factor of diabetic mellitus which is main cause for the stroke. In the introduction, the author described the twos type of stroke like ischemic and hemorrhagic stroke caused by diabetic mellitus and their circumstance’s. As authors described the enhanced generation/accumulation of various advanced glycation end products (AGE) and their corresponding ligands/receptors are responsible for pathomechanism interaction in diabetes-ischemic stroke combination resulting cardiovascular and cerebrovascular diseases.
The cytotoxic and neurotoxicity in diabetic ischemic stroke were found due to the generation of toxic AGEs. The mechanistic aspects of cross linking and non-cross-linking AGEs, toxic substances created by AGEs (TAGEs) and the receptors of AGEs (RAGEs) were well elaborated by the authors with good referencing.
The mechanism associate after the release of HMBG1 and neurovascular mechanism with RAGEs in ischemic stroke was discussed by the authors in the current review.
Clinical findings of therapeutic agents to minimize the strokes were nicely presented by the authors. The drugs which are used for targeting diabetic cardio-cerebrovascular and ischemic stroke were well explored in the review along with their animal model study and with the corresponding summarized outcomes. Although the limited clinical evidence are reported but AGE-RAGE antagonists is still the better therapeutic agent for diabetes-stroke associated complications and metformin therapy was found of having lower risk of ischemic stroke in type 2 diabetes mellitus patients. RAGE, HMGB1, Nitric oxide, Polyamine mechanisms were found as important therapeutic targets in diabetes-stroke associated disorder.
In the later part of the review, authors have elaborated the myokines, neuroinflammation, oxidative stress and neuromuscular electrical stimulation found in ischemic stroke in diabetes mellitus patients.
It is recommended to emphasis the importance of iminosugars and sugar derivatives as an anti-diabetic agents, and it is suggested to cite following relevant articles related to iminosugars in introduction section.
1. Nash, R. J.; Kato, A.; Yu, C-. Y.; Fleet, G. W. J. Iminosugars as therapeutic agents: recent advances and promising trends. Future Med. Chem. 2011, 3, 1513−1521.
2. Yang, L.-F.; Shimadate, Y.; Kato, A.; Li, Y.-X.; Jia, Y.-M.; Fleet, G.W.J.; Yu, C.-Y. Synthesis and glycosidase inhibition of N-substituted derivatives of DIM. Org. Biomol. Chem. 2020, 18, 999–1011.
3. Chennaiah, A.; Dahiya, A.; Dubbu, S.; Vankar, Y. D. A Stereoselective Synthesis of an Imino Glycal: Application in the Synthesis of (−)-1-Epi -Adenophorine and a Homoiminosugar. Eur. J. Org. Chem. 2018, 6574−6581.
4. Chennaiah, A.; Bhowmick, S.; Vankar, Y. D. Conversion of glycals into vicinal-1,2-diazides and 1,2-(or 2,1)-azidoacetates using hypervalent iodine reagents and Me3SiN3. Application in the synthesis of N-glycopeptides, pseudo-trisaccharides and an iminosugar. RSC Adv. 2017, 7, 41755−41762.
5. Rajasekaran, P.; Ande, C.; Vankar, Y. D. Synthesis of (5,6 & 6,6)-oxa-oxa annulated sugars as glycosidase inhibitors from 2-formyl galactal using iodocyclization as a key step. ARKIVOC 2022, vi, 5−23.
Overall, after addressing the points mentioned above, I recommend this review to publish in biomolecules.
Author Response
Dear Reviewer,
Please see the attachment for author responses.

Reviewer 2 Report (New Reviewer)
Reviewer comments and suggestions
The authors in this review discuss the evidence on key roles of AGEs, RAGEs, other ligands- Leukotriene B4 (LTB4), High mobility group box 1 (HMGB1) nuclear protein, Brain-kidney-muscle cross-talks, alternate pathomechanisms in neurodegeneration related to diabetic ischemic stroke. In the literature it has provided evidence of RAGE, HMGB1, Nitric oxide, Polyamine mechanisms may have an important role in therapeutic targets, inflicting common consequences of neuroinflammation and oxidative stress.
Finally, the authors suggested a comprehensive therapeutic strategy that features all the identified components is needed for outcome improvement in diabetic stroke patients.
The manuscript needs a thorough English correction and I could find some structural errors in the sentences that needed to be rectified with the help of English speakers.
A few concerns/comments needed to be explained/modified.
- Line 7-8 The affiliations that the authors mentioned were wrong. please check the author's guidelines
- In line 36 the authors have to mention here about five therapeutics
- Line 57 CNS First time used so should be in full form
- Line 91 Only reference 28 is not enough to discuss the line 91-92
- Line 186 “ Diagnostic studies should exclude potential sources of cardiogenicembolism”. what does it mean
- Line 206-207 Need an appropriate reference
- Line 2130215 You can cite references such as https://www.nature.com/articles/hr201767
https://pubmed.ncbi.nlm.nih.gov/29556093/
https://pubmed.ncbi.nlm.nih.gov/25180150/
8. Table 2 therapeutic agent should be drug as well and the authors could elaborate more contents in the tables, these comments is for both table 3 and 4
9. Comments for table 4 have the authors discussed vitamins in the text; I did not find any portion
10. In the conclusion part, Nitric should be nitric, and polyamines
11. Line 662 If the authors want four other keys,, then they should mention all or otherwise delete it with the appropriate sentence
12. Please modify all the references according to the MDPI journals
Author Response
Dear Reviewer,
Please see the attachment for author responses.

Round 2
Reviewer 2 Report (New Reviewer)
All comments were incorporated in the MS.
This manuscript is a resubmission of an earlier submission. The following is a list of the peer review reports and author responses from that submission.
Round 1
Reviewer 1 Report
The prevalence of diabetes and its associated complications have significantly increased in the past two decades. It has become a major public health burden. The Lack of biomarkers for early diagnosis and effective therapeutics has further worsened the situation. In this manuscript, the authors reviewed the role of RAGE and its ligands in the pathogenesis of diabetic ischemic stroke and discussed the potential of this pathway as a therapeutic target. The topic is very interesting and the authors have nicely reviewed and discussed the relevant literature. The reviewer would like to make a few comments to improve the manuscript
One of the major weaknesses of the manuscript is the superficial review/discussion of the published literature. For example
- Line 186-195: in this section, the authors elaborated on how RAGE mediates stroke. However, the paragraph does not describe the actual mechanism instead it just says RAGE and HMGb1 are mediators of stroke. If there is published literature on the mechanism, then please describe those. If not, then please outline your hypothesis.
- Line 193-194: “It has been shown that RAGE may be involved in a more vigorous inflammatory response mediated by high-mobility group box 1 (HMGB-1) after stroke”. What are these inflammatory responses? Without mentioning those responses, this sentence remains a vague statement and does not add anything to the story.
- How is HMGB1 linked to stroke? Please elaborate on the mechanism.
- Line 202: how does endothelial RAGE contribute to ischemia pathogenesis?
- Line 211: “Decoy receptor soluble RAGE and genetic RAGE deficiency reduced the infarct size” – please add a reference for this statement.
- Line 214: Please describe the relevance of macrophages in this section.
- Line 249-257: How is this section relevant to diabetic ischemic stroke?
- Line 266-269: What is the relevance of ApoE4 in stroke?
- In several sections, the authors used references from Alzheimer’s disease and RAGE. However, it has not been described how stroke and Alzheimer’s disease are linked.
Elaborate abbreviations when first used e.g. CT, MRI in line 56. Please go through the manuscript and edit where necessary.
Line 239-21: Please revise this first sentence – preferably split into two sentences for easy interpretation.
Line 352-363: please elaborate on how this section is relevant to diabetes and its associated ischemic stroke?
The manuscript is well written. However, there are a few grammatical errors. A moderate English editing is required.
Reviewer 2 Report
The authors summarized the receptors for advanced glycation end products, organ crosstalk, and pathomechanism targets for comprehensive molecular therapeutics in diabetic ischemic stroke. It is an important topic.
The manuscript did not provide the information about the levels of advanced glycation end products measured and compared between diabetic ischemic stroke and non-diabetic ischemic stroke, the time course of these AGE products contributing to the severity of the diabetic ischemic stroke, and whether renal dysfunction was derived from diabetes, etc.
Toxic AGEs are listed in Table 1. Are other AGEs not toxic, which are essential in normal cell activities? They should clearly state it.
The authors discussed the RAGE-mediated ischemic brain damage, neurovascular changes, interplay of leukotriene B4 receptor, neurotoxicity, etc. Are they acute effects or chronic effects? What time courses do these reactions occur in stroke patients? Are they specific to diabetic ischemic patients?
The cardio-cerebrovascular complication was mentioned several times in the manuscript. However, the manuscript content does not include the heart. Will the authors add some heart studies?
The title page has the note "deemed to be university." It is not necessary and should be removed.
Reviewer 3 Report
Enhanced generation, accumulation of advanced glycation end products (AGEs) occurring due to chronic hyperglycemia of diabetes has been implicated in pathogenesis of stroke. The review presents current and previous findings on AGEs, and neurotoxicity in diabetic ischemic stroke, obtained from human subjects and animal models to decipher the intricate interplay of molecular mechanisms. This will explicate the close interaction diabetes mellitus, and neurotoxicity due to ischemic stroke.
However, the manuscript has to be amended to be accepted in the journal as review article.
#1. I would like the authors to focus on the link between AGE or RAGE and ischemic stroke in patients with diabetes. Several parts do not indicate such information, for example, in page 7, line 248-257, and page 8, line 310-322. Please omit descriptions that do not include the information.
#2. In Table 2, too many agents are indicated, but the authors have to cite each original article appropriately showing the results and what models (e.g., in vitro, in vivo, name of model mice or rats, name of cultured cells) were used. Table 2 does not demonstrate which agents are ready to use for inhibition of the pathway after RCTs or still in progress to be established in the clinical setting.
#3. In Table 2, are polyamine and nitric oxide synthase associated with inhibition of AGE-RAGE? Ref. 22 does not include such description.
#4. In Abstract, the authors mention that stroke and diabetes are two leading causes of morbidity and mortality. According to WHO, stroke is a second cause of mortality worldwide, but diabetes mellitus is a ninth cause. Therefore, I think the authors should describe diabetes mellitus is a ninth cause of mortality, not a leading cause of mortality.
#5. In page 2 and line 55, the definition of stroke, the authors explain that it is focal neurological signs or symptoms thought to be of vascular 55 origin that persist for >24 h confirmed by brain CT and/or MRI. However, the definition above is clinically defined. Ref. 5 indicates two patterns of the definition, one is a definition showing pathological, imaging, or other objective evidence of cerebral, spinal cord, or retinal focal ischemic injury in a defined vascular distribution, and the other is clinical evidence indicated in the manuscript. Therefore, the authors should introduce the two definitions.
#6. In page 2, line 60-62 is a well-known fact. The authors do not need to describe it.
#7. In page 4, line 155, the authors say the main cause of stroke is atherosclerosis. What subtypes of stroke do the authors mean? Ischemic stroke is formally classified according to the TOAST classification. The authors have to describe atherosclerosis according to the TOAST classification. What subtypes in the TOAST classification is diabetes mellitus associated with?
#8. Add appropriate citations in line 177, 185, 246
#9. In page 6, line 221-222, BBB includes endothelial cells as one of the components. Precisely indicate the sentences.
#10. In page 7, line 267, the authors wrote Apo E4, and have to write APOE4 or ApoE4 not Apo E4. Is the Apo E4 a murine or human gene? The former is human gene, the latter murine gene.
#11. In page 8, line 303, the authors describe effects of TAGE in Schwann cells, which are located in white matter. However, in ‘6. Neurotoxicity of AGEs demonstrated from animal and human models’, functions of glial cells and neurons are explained. Neurons are located in grey matter. The authors must explain effects of neurotoxicity of AGEs in territories of grey matter and white matter respectively. Grey matter and white matter are totally different in structure and function.
#12. In page 13, line 428-436, the authors should indicate the direct relationship between AGE or RAGE and quercetin in advance. The manuscript does not show the direct relationship in the main text. As I mentioned above, it is not clear a relationship between AGE or RAGE and polyamine. It is natural that antioxidant therapies will combat oxidative stress under hyperglycemia. The authors raise specific candidates of antioxidants.
#13. In page 13, line 440, the authors do not show a direct relationship between AGE or RAGE and myokine.
Round 2
Reviewer 1 Report
Dear Authors
Thank you for the revised version of the manuscript. I am still struggling to understand the take-home message of a few sections. For example
In response 1, the authors described the mechanism mediated by RAGE and HMGB1. Paragraphs 1, 2 and 3 described important information but they are disconnected and not appropriately blended. This makes it difficult to follow.
Response 4: the authors did not mention any details on the endothelial RAGE-mediated derangement of microvascular circulation.
Response 6: "Data from animal model experiments using chimeric mice, both in vivo and in vitro" - in this paper, the authors used chimeric mice only in in-vivo experiments. Another sentence in this paragraph "Further enhancement in HMGB1 toxic effect was caused by macrophage addition to neural cultures." This is basically the repeat of the newly added text.
Response 8 created more confusion. It is not clear how AGE-RAGE system is linked to APOE4?
Now there are more short paragraphs containing 2-4 sentences i.e. those paragraphs lack appropriate structure.
In the responses, the line numbers do not match the main text which made it difficult to follow the changes.
Kind Regards
Reviewer 2 Report
My concerns have been resolved
Reviewer 3 Report
I think the authors are familiar with molecular science regarding AGE and RAGE well. However, they do not have a detailed knowledge about neurology and neuroscience. They have several wrong explanations.
(1) Diabetes mellitus is associated with only small vessel disease among all subtypes of ischemic stroke according to TOAST classification in the revised manuscript, but indeed diabetes mellitus is a main cause of large-artery atherosclerosis as well. Furthermore, Ref.8 does not indicate that DM is associated with SVO in the Results.
(2) Neurovascular unit consists of endothelial cells and blood-brain barriers in the revised manuscript, but components of blood-brain barriers include endothelial cells. Astrocytes, and pericytes are involved in neurovascular unit.
(3) The authors indicate that ischemia-induced Aβ/tau pathology similar to Alzheimer’s disease is reported to be involved in post-stroke cognitive impairment in line 318-319 in the revised manuscript. That is correct. However, this context and Ref. 36 do not refer to post-stroke cognitive impairment but vascular cognitive impairment. The authors should have knowledge the difference between them.
In line 212, Ref. 8 does not show diabetes may cause more subtle brain damage indicated by lacunar infarcts, which raises the risk of dementia and causes a faster loss in cognitive function.
In line 252-265, the descriptions have nothing to do with AGE-RAGE relationship.
In line 335-336, AGE-RAGE system may also participate in Apolipoprotein E4 (APOE4)-associated pathological changes. Please add a citation.
In line 384-385, Impaired vasodilation in diabetes may be a result of reduction of NO activity by AGEs. Please add a citation.
There are some typos (eg, Nitric oxide in line 378 have to be begun with small letter n.).